# SARS-CoV-2 Omicron (B.1.1.529) Variant: A Challenge with COVID-19

**DOI:** 10.3390/diagnostics13030559

**Published:** 2023-02-02

**Authors:** Zeinab Mohseni Afshar, Ali Tavakoli Pirzaman, Bardia Karim, Shiva Rahimipour Anaraki, Rezvan Hosseinzadeh, Elaheh Sanjari Pireivatlou, Arefeh Babazadeh, Dariush Hosseinzadeh, Seyed Rouhollah Miri, Terence T. Sio, Mark J. M. Sullman, Mohammad Barary, Soheil Ebrahimpour

**Affiliations:** 1Clinical Research Development Center, Imam Reza Hospital, Kermanshah University of Medical Sciences, Kermanshah 51351, Iran; 2Student Research Committee, Babol University of Medical Sciences, Babol 47176-47745, Iran; 3Faculty of Medicine, Iran University of Medical Sciences, Tehran 14535, Iran; 4Student Research Committee, Faculty of Pharmacy, Islamic Azad University, Ayatollah Amoli Branch, Amol 4615143358, Iran; 5Infectious Diseases and Tropical Medicine Research Center, Health Research Institute, Babol University of Medical Sciences, Babol 47176-47745, Iran; 6Bogomolets National Medical University, 01601 Kyiv, Ukraine; 7Cancer Research Center, Canicer Institute of Iran, Tehran University of Medical Science, Tehran 1419733151, Iran; 8Department of Radiation Oncology, Mayo Clinic, Phoenix, AZ 85054, USA; 9Department of Social Sciences, University of Nicosia, Nicosia 2417, Cyprus; 10Department of Life and Health Sciences, University of Nicosia, Nicosia 2417, Cyprus; 11Student Research Committee, Virtual School of Medical Education and Management, Shahid Beheshti University of Medical Sciences, Tehran 19839 69411, Iran

**Keywords:** omicron, COVID-19, SARS-CoV-2, variants of concern

## Abstract

Since the beginning of the coronavirus disease 2019 (COVID-19) pandemic, there have been multiple peaks of the SARS-CoV-2 (severe acute respiratory syndrome coronavirus virus 2) infection, mainly due to the emergence of new variants, each with a new set of mutations in the viral genome, which have led to changes in the pathogenicity, transmissibility, and morbidity. The Omicron variant is the most recent variant of concern (VOC) to emerge and was recognized by the World Health Organization (WHO) on 26 November 2021. The Omicron lineage is phylogenetically distinct from earlier variants, including the previously dominant Delta SARS-CoV-2 variant. The reverse transcription–polymerase chain reaction (RT–PCR) test, rapid antigen assays, and chest computed tomography (CT) scans can help diagnose the Omicron variant. Furthermore, many agents are expected to have therapeutic benefits for those infected with the Omicron variant, including TriSb92, molnupiravir, nirmatrelvir, and their combination, corticosteroids, and interleukin-6 (IL-6) receptor blockers. Despite being milder than previous variants, the Omicron variant threatens many lives, particularly among the unvaccinated, due to its higher transmissibility, pathogenicity, and infectivity. Mounting evidence has reported the most common clinical manifestations of the Omicron variant to be fever, runny nose, sore throat, severe headache, and fatigue. This review summarizes the essential features of the Omicron variant, including its history, genome, transmissibility, clinical manifestations, diagnosis, management, and the effectiveness of existing vaccines against this VOC.

## 1. Introduction

The SARS-CoV-2 virus, which has been prevalent worldwide for almost three years, has caused the death of more than 6 million people and infected more than 500 million people with COVID-19. It has a fragile possibility of elimination and is most expected to circulate endemically around the world [1,2,3]. Despite such an expectation, the emergence of new variants that spread rapidly in countries and geographical regions threatens the predicted change to endemism for this virus [4,5]. since the beginning of the SARS-CoV-2 pandemic, the WHO has declared 5 variants of concern (VOCs), which are known as Alpha, Beta, Gamma, Delta, and Omicron [6,7]. Changes in the viral genome can make these new variants more transmissible, lethal, and harder to treat. The most recent SARS-CoV-2 variant, Omicron, has raised significant concern worldwide [8]. In the current paper, we have provided a summary of the Omicron variant and its important features like history, genome, transmissibility, clinical manifestations, diagnosis, management, and the effectiveness of the existing vaccines against this VOC.

## 2. History

The COVID-19 outbreak was identified in December 2019 [9]. Since this time, multiple peaks of the SARS-CoV-2 infection have emerged, mainly due to the emergence of new variants, including the Alpha (B.1.1.7), Beta (B.1.351), Gamma (P.1), Epsilon (B.1.427 and B.1.429), Delta (B.1.617.2), Mu (B.1.621), and Lambda (C.37) variants, each with a new set of mutations in the viral genome, leading to different pathogenicity, transmissibility, and morbidity [10]. The Omicron (B.1.1.529) variant is the most recent VOC to emerge and was recognized by the WHO on 26 November 2021, from a sample collected on 9 November 2021 [11,12]. This variant was first reported in Botswana and South Africa but quickly spread to other countries [13].

## 3. Genome

The Omicron lineage has been demonstrated to be phylogenetically distinct from the previous variants, including the previously dominant Delta SARS-CoV-2 variant [14]. The Omicron variant has experienced a total of 18,261 mutations in its genome, of which only 588 mutations were in the extragenic region, most of which occurred in the coding region. Among the mutations in the coding region, 2743 were synonymous single-nucleotide polymorphisms (SNPs) mutations, and 11,995 were non-synonymous [15,16].

The Omicron variant evolved with 37 amino acid substitutions from SARS-CoV-2 spike protein, several in the receptor-binding domain (RBD) [17] (Figure 1). According to a recent study, Omicron-BA.1 has undergone 50 mutations and has 34 changes in its spike protein gene, 15 of which occurred in the RBD [16,18]. Thirty single-point substitutions (including A67V, T95I, G142D, L212I, G339D, S371L, S373P, S375F, K417N, N440K, G446S, S477N, T478K, E484A, Q493K, G496S, Q498R, N501Y, Y505H, T547K, D614G, H655Y, N679K, P681H, N764K, D796Y, N856K, Q954H, N969K, and L981F), three deletions (including Δ69–70, Δ143–145, and Δ211) and one insertion (ins214EPE) have been identified on the spike protein of the Omicron variant [19]. Several investigations in South Africa have revealed that D614G, N501Y, K417N, T478K (concerning the mutations), and some new mutations in this VOC are responsible for its relative resistance to the current vaccines and the enhancement of its reinfection rate [20]. ins214EPE is also a mutation with the insertion of three amino acids in Omicron-BA.1, which can be characteristic, but its role has not yet been determined [21]. D614G, E484K, K417N, T478K, and N501Y are substantial mutations in the RBD and have also been identified in previous variants of SARS-CoV-2. Based on previous studies, they can increase the overall risk of reinfection and relative resistance to existing vaccines [22]. E484K, a glutamic acid to lysine substitution at position 484, is a significant mutation that has also been detected in both the Beta and Gamma variants [23]. It has been hypothesized that this mutation led to the enhanced reinfection rate found in the Gamma variant [24].

However, E484A, as the counterpart mutation, is a glutamic acid to alanine substitution at position 484, found in the Omicron variant. The mutation of glutamic acid (a hydrophilic amino acid) to alanine (a hydrophobic amino acid) might have the ability to change the interaction between the angiotensin-converting enzyme 2 (ACE2) and the RBD [23]. Interestingly, two of the three RBD mutations in the Omicron variant are shared with a previous VOC, the Delta variant. The first mutation is a lysine-to-asparagine exchange at position 417, which results in structural changes to the S protein and is probably responsible for the variant’s enhanced ability to escape the immune system. The second one, a threonine-to-lysine exchange at position 478, has probably ameliorated the residue’s electrostatic potential and steric interference. Therefore, it has been linked to elevated RBD binding affinity and improved immunological evasion [25]. However, the mutation in the Delta type by substituting leucine instead of arginine at position 452 strengthens the affinity of this variant to ACE2 receptors found in various human cells, including the lung, which is not present in the omicron variant [25]. The Omicron variant consists of several sublineages, including BA.1 (known as the original Omicron), BA.1.1, BA.2, BA.2.12.1, BA.2.3, BA.2.9, BA.3, BA.4, and BA.5 [26,27]. Besides, several newly-emerging sublineages have been detected like BQ.1, BQ.1.1, BA.4.6, BF.7, and BA.2.75.2. Importantly, it has been reported that all new subvariants had enhanced neutralization resistance, especially BA.2.75.2, BQ.1, and BQ.1.1. The improved neutralization resistance is attributed to the mutations in these variants, which are explained in the following section [28]. The Omicron variant has the most significant number of mutations of the known SARS-CoV-2 variants, which has made this variant more able to avoid the neutralizing antibodies induced by natural infection or vaccination [29].

Among other mutations that increase the infectivity in the Omicron variant, we can mention the mutations created in the virus nucleocapsid gene, that the double substitution mutation R203K + G204R observed in Alpha, Gamma, and Omicron variants is one of the prominent mutations of this gene. In a study that has investigated the effect of this mutation, it has been seen that the occurrence of this mutation has increased the proliferation of the virus by increasing the phosphorylation of the nucleocapsid and based on the observations they have had in vitro and in vivo, they state that the occurrence of such mutations in genes outside the spike can increase the compatibility of the virus and its infectivity. Thus, it is imperative to pay attention to these mutations [30].

## 4. Sublineages of the Omicron Variant

Globally, a total number of 552,191 confirmed cases and 115 deaths were reported, by 8 January 2022, in 150 countries. The United States included 62,480 cases and only 1 confirmed death [31]. The Omicron VOC is the variant spreading worldwide, causing almost all sequences reported to GISAID. Although these sublineages have a wide range of genetic diversity, causing various mechanisms of immune escape, they share equivalent clinical outcomes [32]. In addition to sublineages like BA.1 (known as the original Omicron), BA.1.1, BA.2, BA.2.12.1, BA.2.3, BA.2.9, BA.3, BA.4, and BA.5, several concerning subvariant have emerged, demonstrating features like enhanced reinfection risk and immune escape. Currently, no epidemiological data indicate an increase in the disease severity caused by these variants [32].

Previously, BA.2 displaced the original Omicron BA.1. However, it continued to evolve to new subvariants like BA.2.12.1, BA.2.75, BA.2.75.2, BA.4, and BA.5. In many countries, BA.5 is now the dominant subvariant. Notably, the descendants of BA.4 and BA.5, such as BQ.1.1, BF.7, and BA.4.6, are growing in prevalence. In the United States, BQ.1, BQ.1.1, BF.7, and BA.4.6 caused 25.5%, 24.2%, 7.8%, and 4.4% of all cases, respectively, by 19 November 2022. XBB is another descendant of the BA.5 subvariant detected in India in August 2022 for the first time. This sublineage is growing in prevalence in Europe and is also identified in the United States. From 3 October to 9 October 2022, it caused 54% of COVID-19 cases in Singapore [33].

Descendant Omicron subvariants include various amino acid sequences in the critical parts of their genome, such as S protein and nucleocapsid proteins. S protein, the most crucial protein playing a pivotal role in viral entry and pathogenicity, contains 31–37 mutations. Descendant Omicron subvariants share many of these mutations among themselves: P681H, N679K, H655Y, and D614G in the S1 subunit; N969K, Q954H, D796Y, and N764K in the S2 subunit; Y505H, N501Y, Q498R, E484A, T478K, S477N, N440K, K417N, S375F, S373P, S371L/F, and G339D in RBD; and G142D in N-terminal domain (NTD) (Figure 2). Some of these mutations, especially NTD mutations, could enhance the evasion of viruses from NTD-targeted neutralizing antibodies [26,34]. Figure 2 provides a better and simpler view of these mutations and their function in each sublineage.

A subvariant of BA.5 is BQ.1 variant. It has been identified in 65 countries and was responsible for a 6% prevalence. It contains spike mutations like N460K and K444T in some vital antigenic sites. Moreover, the BQ.1.1 subvariant contains a fundamental mutation in a vital spot, like R346T. Due to a lack of data, it is impossible to determine these new variants’ immune escape or severity precisely. However, the enhanced growth advantage of these variants compared to other subvariants warrants precise and close monitoring and investigations [32]. Another important sublineage is BA.2.75.2, derived from BA.2 and identified in Singapore and India at first. It contains several mutations, including D1199N, F486S, and R346T [36]. Notably, it has been reported that the R346T mutation was associated with enhanced evasion from vaccine-induced and monoclonal antibodies [37].

## 5. Transmissibility and Infectivity

Binding affinity with ACE2 complex and RBD in SARS-CoV-2 viruses plays a significant role in determining their binding affinity, but furin cleavage sites also play an essential role in this field [38,39]. According to this issue, it is evident that mutations in the viral genome can increase the affinity of the virus to the host cells and lead to higher transmissibility. This has been experienced previously, as the Alpha, Beta, and Delta variants have had 7-, 19-, and 11-times higher transmissibility than the original SARS-CoV-2 virus [40,41,42]. As some of the mutations of the Omicron variant are mapped on the receptor-binding motif, its spike protein affinity towards the ACE2 receptor is much higher than the previous variants [43]. Also, based on the studies conducted, Omicron shows a significant change in its infectivity due to three mutations in the cutting site of furin and 15 mutations in RBD [38,39,44]. The ACE, coded on the RBD, is the main gate of viral entry into human cells. Therefore, 2- to 3-times increased transmissibility than the Delta variant would be expected [45,46]. In a study that examined the infectivity of Omicron, they observed that most of the RBD mutations, except for the G339D, S371L, S373P, and S375F mutations, were created close to the ACE2 and RBD binding interface. As a result of these mutations, changes in binding free energy are significantly increased, making Omicron more infectious due to the increased binding affinity of the ACE2-RBD complex [44].

Furthermore, the shorter doubling time of the Omicron variant and the higher viral load induced in the nasopharyngeal and respiratory cells, compared with previous VOCs, also confirm its higher infectivity [47,48]. It has been demonstrated that the infectivity of the Omicron variant is approximately ten times higher than the wild-type strain [49]. It should be stated that, among all the possible factors, mutations have one of the most significant impacts on the high transmissibility of this VOC. The N501Y is one of the most critical mutations that can enhance the binding affinity with the ACE2 receptor, thus increasing transmission. In association with Q498R, the N501Y mutation can strengthen the binding affinity and make it easier for the Omicron variant to enter the host cells [20]. A recent study detected two subclades within the Omicron lineage, with K417N or K440N mutations and S446K. Subsequently, it has been mentioned that the K417N mutation, found in the Beta variant, can also moderately enhance the surface expression of the RBD and increase resistance to the neutralizing monoclonal antibodies [50].

Furthermore, several studies have suggested that the furin cleavage site (FCS), located in the SARS-CoV-2 spike protein, boosts RBD exposure and its binding to the ACE2 receptor [51,52]. Mutations like H655Y and N679K, located close to the FCS, can enhance spike cleavage and make the Omicron variant more transmissible [20]. According to recent studies, the H655Y mutation, detected in the Gamma and Omicron variants, was accompanied by antigenicity alterations, which enhanced monoclonal antibodies’ evasion [53].

## 6. Clinical Manifestations

It is believed that the incubation period of the Omicron variant is shorter than previous SARS-CoV-2 variants, with a median of three days, compared with at least four days for previous strains [54,55]. The presentations of the Omicron variant were expected to be the same as with other variants. However, reviewing the available literature has revealed that fever, runny nose, sore throat, severe headache, and fatigue are the predominant clinical manifestations of this variant [56,57,58]. Among the two types of Omicron, based on the available evidence, the severity of the disease in subtype BA2 is higher than its severity in B.1 [59]. Most reported Omicron cases have been mildly affected, especially those previously infected or vaccinated [60,61]. Moreover, young and middle-aged individuals are more commonly infected with this variant than previous variants [62], as reflected in the rapid increase of pediatric admissions, due to SARS-CoV-2 infection, during the early days of the Omicron wave in South Africa [63]. Fortunately, most cases of Omicron do not require hospitalization or intensive care unit (ICU) admission [64].

It is not yet understood whether the mild features of this variant are due to the attenuated nature of the virus or the existing immunity among those infected. However, some studies have concluded that the lower severity of the Omicron-induced infection could be due to its slower replication in the transmembrane serine protease 2 (TMPRSS2) than in previous variants [65]. The underlying reason might be that the TMPRSS2, a necessary component for activating the spike protein during membrane fusion, plays a less critical role in the Omicron variant [66]. Therefore, despite faster replication in the bronchus, it is believed that omicron replication is slower in the lung parenchyma compared with the previous strains, such as the Delta variant [67].

## 7. Disease Severity

The available data in this area is incomplete. However, based on preliminary results in South Africa, this virus shows a lower hospitalization rate than previous infections caused by the Delta variant. Also, according to the announcement of the insurance company Discovery Health, people with Omicron have a 29% lower risk of hospitalization compared to the previous variant. Although this information can indicate that the severity of omicron infection is milder than in previous variants, it is still too early to conclude. Various factors can cause disturbances in these statistics, including the patient’s previous exposure to the coronavirus and their age. It has also been confirmed that in South Africa, more than 70% of the population of Omicron-infected areas have already been exposed to SARS-CoV-2, and this point, along with the 40% statistics of injecting at least one dose of the COVID-19 vaccine in This population can affect the severity of the disease in them. These results are in contrast to the results of the Imperial College of London, which did not show any reduction in the hospitalization rate of Omicron patients compared to the Delta variant, and such a result was also seen in a study conducted in Denmark. However, both of these studies are not very reliable since they did not examine a large number of people. However, some studies have shown that, unlike the previous types, the severity of the disease was less for patients with the Omicron variant. This has been demonstrated in shorter hospitalization, lower supplemental oxygen requirements, fewer ICU admissions, and lower mortality [68,69,70,71,72,73].

## 8. Diagnosis

Molecular tests, namely the reverse transcription-polymerase chain reaction (RT-PCR), have been the main laboratory-based diagnostic tests for detecting SARS-CoV-2 throughout the pandemic [74]. Nevertheless, as RT-PCR assays target the spike gene, and because the available RT-PCR assays cannot detect all target genes, it is possible that their failure rate in detecting the new variants, including Omicron, will be higher than for previous strains [75]. The sensitivity of the RT-PCR tests for diagnosing the original SARS-CoV-2 has been estimated to be up to 60–70%, depending upon the stage and severity of the infection and the accuracy of the nucleic acid detection technique [76]. Nonetheless, it is reasonable to assume that new mutations in the viral spike protein and RBD can lead to decreased sensitivity of the molecular diagnostic methods. Moreover, serology assays may also have lower sensitivity based on the SNPs of the early strain S protein [77]. However, despite the mutations mentioned above, using highly conserved domains of the SARS-CoV-2 genome as the RT-PCR targets, and performing variant-specific RT-PCR tests, can add to the sensitivity and reduce test failure [78,79].

Furthermore, specific rapid antigen assays have been shown to detect Omicron more accurately [80]. In addition, collecting saliva swab specimens instead of mid-turbinate swabs might further increase the sensitivity of these assays [81]. The abovementioned strategies could improve the timely diagnosis of patients and, as a result, the effective interruption of the transmission chain. Moreover, CT chest scans are highly sensitive in detecting SARS-CoV-2 infection in previous variants [82], and the imaging findings of Omicron-induced pneumonia are no different. It is believed that the CT scan findings of those infected with Omicron are consistent with minimal to mild pneumonia [83].

## 9. Management

The main route SARS-CoV-2 uses to enter the host cells is via spike glycoprotein attachment to the ACE2 receptors, so any variation in the virus’s genome can reduce therapeutics’ effectiveness, which aims to inhibit viral attachment. Therefore, the resistance of this variant to current therapeutics, including monoclonal antibodies (mABs), has been predicted [84]. In other words, as reported by a recent study, 7 out of 9 monoclonal antibodies (including bamlanvimab, etesevimab, casirivimab (REGN10933), imdevimab (REGN10987), sotrovimab (S309), DZIF-10c, P2B-2F6, C102, and Fab2-36) could not demonstrate efficient neutralizing activity against the Omicron variant. However, these effectively neutralized the Wu01 strain and the Alpha variant. Interestingly, the Delta and Beta variants showed partial resistance to these monoclonal antibodies, with 7 of 9 and 5 of 9 demonstrating sufficient neutralizing activity against the Delta and Beta variants, respectively [85]. However, many agents are predicted to have therapeutic benefits on the Omicron variant. For example, TriSb92, a trimeric human nephrocystin SH3 domain-derived antibody, is believed to inhibit the new variant if administered intranasally [86].

Furthermore, a group of recent studies has reported that some antivirals, including molnupiravir, nirmatrelvir, and their combination, could significantly prevent infection with Omicron and previous VOCs [87,88,89,90]. According to an update on the Omicron variant by WHO, corticosteroids and IL6 receptor blockers can still manage severe COVID-19 cases [91]. Also, one of the treatment options that doctors welcomed was the use of convalescent plasma for the treatment of COVID-19 disease, which means the use of antibodies in the plasma of convalescing patients to fight infection in patients receiving plasma [92]. The clinical trial conducted on patients with COVID-19, SARS, MERS and influenza showed positive results, but a study that examined the clinical trials and cohort studies did not report many relationships between this treatment and the improvement of results [93,94]. Finally, a clinical trial study that examined the progress of the disease in patients treated with plasma during convalescence did not show any effect on the results of the patients. As a result, after the investigations carried out by the WHO, due to the lack of convincing evidence based on the therapeutic results of convalescent plasma on COVID-19, it is not recommended to use this method to treat hospitalized patients except for a clinical trial framework [95,96]. Given the above, considering the novelty of the Omicron variant, more research is needed to better understand the management options for this variant.

## 10. Effectiveness of Neutralizing Antibodies and Vaccines

Previous studies have shown that the Beta and Delta variants could evade convalescent serum and neutralizing antibodies, leading to a higher risk of reinfection than the wild-type and Alpha variants [97,98]. Nonetheless, this issue is not well understood for the Omicron variant, with laboratory-based neutralization studies currently underway. However, clinical experiences confirm the immune evasion of this variant, as reinfections are being reported in individuals previously infected with other variants [8]. Moreover, due to the increased mutations in the RBD, and since the spike’s RBD is the principal target for neutralizing antibodies, the Omicron variant is expected to be neutralized less effectively by antibodies and vaccination than the previous variants, including the Delta variant [99]. Therefore, as some mABs are being used as therapeutics for those infected with SARS-CoV-2, it would seem likely that the Omicron variant will be resistant to some of the current treatment strategies, including most mABs [100]. Studies have demonstrated that a combination treatment of casirivimab/imdevimab (sold under the brand name REGEN-COV) cannot effectively neutralize the Omicron variant [101]. Nevertheless, despite their less beneficial effect, broadly neutralizing mABs with more conserved genome targets may be helpful against the Omicron strain [84].

Vaccination is considered the most effective means of preventing and controlling COVID-19, and four types of vaccines have been introduced for this disease, which include viral vaccines, viral vector vaccines, DNA/RNA vaccines, and protein-based vaccines. Since the target of the current COVID-19 vaccines is the S protein of these viruses and due to the changes that have occurred in the spike protein of the Omicron variant, the ability of the variant to escape from the current vaccines may have significantly increased [102,103,104]. Similar to the Beta and Delta variants, the neutralization efficiency of the COVID-19 vaccines against the omicron variant is considerably lower than for the wild-type [105,106]. The plasma of individuals who had received two mRNA vaccine doses had several times less potency against the Omicron variant than for the original strain [99]. It is predicted that the vaccine-escape ability of the Omicron variant is twice that of the Delta variant [107]. All these hypotheses have been confirmed because many of the Omicron variant-infected patients had already been fully vaccinated, proving the immune evasion of the Omicron variant [108].

Nonetheless, despite the reduced efficacy of current COVID-19 vaccines against the new variant, they have decreased severe disease, hospitalization, and mortality [69]. Previously vaccinated individuals are expected to develop less severe illnesses if infected with Omicron [109]. Moreover, it has been demonstrated that those with a history of SARS-CoV-2 infection and two vaccine doses or without a history of infection but who have received three vaccine doses have comparable immunity against these new variants and the wild-type virus [110,111,112].

A study has observed that even though Omicron has reduced the effectiveness of the Pfizer-BioNTech vaccine, it can still reduce the risk of hospitalization. Also, Pfizer-BioNTech has stated that despite the mutations in the spike protein of this variant, two doses of these vaccines still protect the patient against severe disease because the T cells created after vaccination are not affected by these Omicron mutations [113,114]. This reflects the synergistic effect of elevated antibody levels after being repeatedly exposed to the antigen, and the impact of affinity maturation [85,115], further justifying the importance of the third vaccine dose. This is while the results of computer modeling show the ability of B.1.1.529 to prevent the development of immunity by T cells [116]. Meanwhile, none of the recipients of the Coronavac vaccine had detectable antibodies [20]. Also, in a study conducted by the most significant private health insurance company in South Africa, the vaccine’s effectiveness against Omicron was reported to be 33%, while for the Delta variant, the vaccine’s effectiveness was estimated to be 80% [113]. Also, based on another study conducted in South Africa, the serum levels of people who had injected the Pfizer-BioNTech vaccine showed 40 times less resistance to the Omicron variant than to the Delta variant [117]. Considering that the ability to neutralize the omicron type in the recipients of the mRNA type vaccine was reduced 4–6 times compared to the wild type, the received vaccine may protect people from contracting the severe type of the disease [20,21]. Therefore, various companies producing COVID-19 vaccines have started studying the production of vaccines focused on the Omicron type.

## 11. Mortality and Prognosis

Even though the previous VOCs, including Alpha, Beta, Gamma, and Delta variants, resulted in a high rate of mortality worldwide [118], this has not been true for the Omicron variant [119]. In the cohort comparison between the Delta variant and Omicron, it has been observed that the omicron variant has caused less mortality than the Delta, and also the hospitalization rate and other factors related to the poor prognosis of the disease have also been seen less in the omicron variant [120]. However, it is essential to note the forthcoming triple respiratory virus threat, which consists of seasonal influenza and the Delta and Omicron variants, which might increase the mortality rate [121].

## 12. Prevention

As Omicron can transmit more readily and rapidly, and since current therapeutics are expected to be less effective against this new variant, it is vital to take strict measures to prevent the spread of the virus. Since the beginning of the spread of the coronavirus, safety instructions have been specified by the WHO, which include measures such as wearing a face mask, improving ventilation, maintaining social distancing, washing hands frequently, avoiding shake-hands, avoiding touching the face with unwashed hands, travel restrictions, and appropriate isolation and quarantine [122]. Since people’s preventive behavior is closely related to their attitude and perception of risk, and every behavior in a person is based on personal experience, people’s awareness of the impact of preventive behavior in preventing the spread of infection is fundamental [123,124]. For example, hand washing, one of the most cost-effective ways to prevent the infection of COVID-19 and according to the instructions, should be done for at least 40 s with soap or 20 s with alcohol gel when returning home [125,126]. According to the evaluation done in a study, 14% of the participants disinfected their hands less than the prescribed time. Also, 7% of the participants stated that they did not wash their hands when entering the house or did not follow the mentioned method [127].

For this reason, advertisements and public health promotion activities should consider programs to increase public awareness of preventive behaviors and their effectiveness. Considering the higher death rate of people at older ages (people over 50 years old) and especially the elderly who are at greater risk, it is better to stay at home as much as possible and limit your contact with other people outside the family [128,129]. Also, the government should consider processes to increase the protection of the elderly and reduce their risk of infection. Infants and children are infected with COVID-19 like other ages, but the protective rules seem sufficient for them due to the milder symptoms [129]. In the conducted studies, women’s adherence to preventive behaviors has been higher than men’s, and the level of education has also been found to be highly related to the average score of people [124,130,131]. Among other issues that have increased preventive behavior in people is their income level, so governments must determine measures to improve the economic status of weak people [124]. Among other preventive measures that have been implemented, there are measures following the international travel of people. For this purpose, many airports set up screening stations. The purpose of such measures is to identify infected cases and prevent the spread of COVID-19 infection at the regional and global levels [129,132].

Moreover, improving diagnostic methods to detect and treat infected individuals quickly can further diminish the transmission chain, all of which are the cornerstones of infection control [11,47,133]. In addition, considering the relatively long time since the start of the two-dose vaccination program in most countries, adding a booster dose can help to reduce the risk of spreading the new variant [134,135]. Accordingly, some countries have planned more vigilant vaccination programs for their populations to prevent another outbreak. For example, the United States has mandated that all individuals aged 5 years and above receive at least two vaccine doses and that high-risk people should get a third vaccine dose [11]. Since the Omicron variant is expected to be the dominant SARS-CoV-2 strain worldwide, developing vaccines tailored explicitly to the Omicron variant, such as multivalent vaccine strategies, is mandated. Moreover, prioritizing vaccinating individuals at higher risk of severe disease and complications is highly recommended [136].

## 13. Conclusions

Despite being milder than previous types and having a shorter incubation period, Omicron threatens many lives, especially among previously unvaccinated individuals, due to its higher transmissibility, pathogenicity, and infectivity. The clinical manifestations of this disease generally include fever, runny nose, sore throat, severe headache, and fatigue, and these manifestations are primarily mild in people who have been infected or vaccinated. The remarkable thing about this variant is that more young and middle-aged people are affected than the previous types. Although the incidence of this variant is high, most cases do not require hospitalization or admission to the ICU. However, specific strategies, such as using Omicron-targeted drugs and vaccines, are needed to prevent the spread of this type. In addition, implementing preventive measures other than vaccination should also be considered.

## Figures and Tables

**Figure 1 diagnostics-13-00559-f001:**
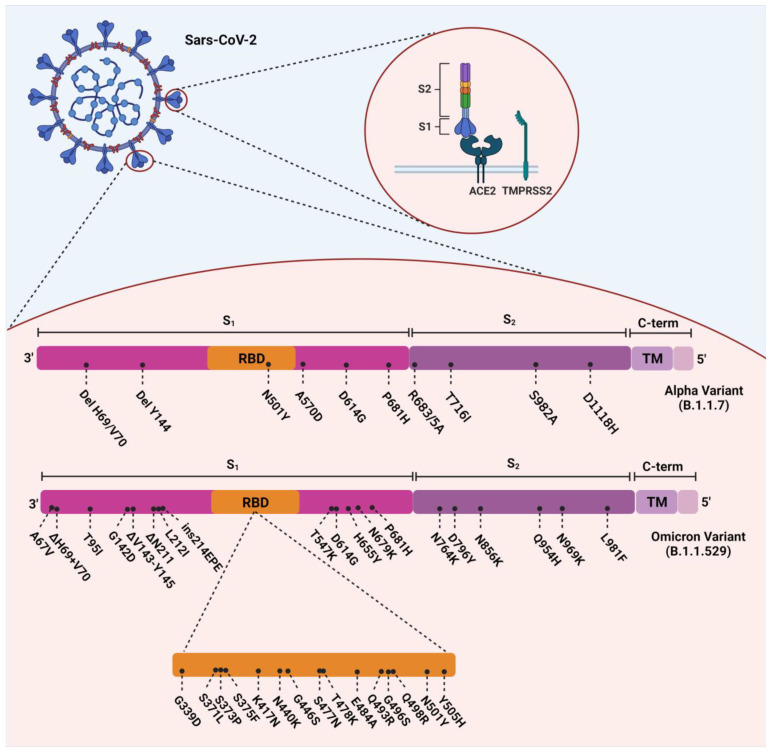
Overview of the mutations of SARS-CoV-2 Alpha (B.1.1.7) and Omicron (B.1.1.529) variants.

**Figure 2 diagnostics-13-00559-f002:**
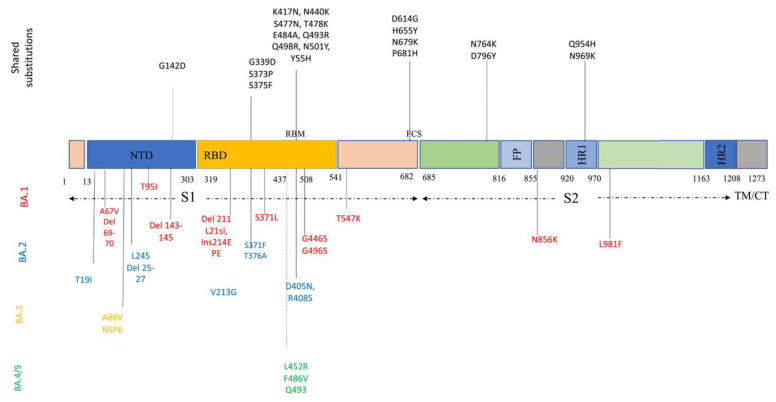
**Amino acid substitutions within the Omicron variant lineage.** Black color represents shared mutations, red Omicron BA.1, blue BA.2, orange BA.3, and green BA.4/BA.5. BA.4 and BA.5 share a similar spike profile as BA.2, except for additional mutations 69–70del, L452R, F486V (green) and reversions to wild type Q493 (Q493R in BA.1, BA.2, and BA.3). BA.4 and BA.5 differ from each other by three amino acid mutations outside Spike. BA.4 additional mutations: ORF7b:L11F, N:P151S. BA.5 additional mutations: M:D3N [35].

## Data Availability

The data that support the findings of this study are available from the corresponding author upon reasonable request.

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
