# Peer review of "SARS-CoV-2 Omicron (B.1.1.529) Variant: A Challenge with COVID-19"

_diagnostics, 2023, doi:10.3390/diagnostics13030559_

Round 1
Reviewer 1 Report (Previous Reviewer 1)
The authors responded and revised the manuscript properly. No more comment is added.
Author Response
Dear reviewer, thank you for your comments and suggestions, which help us significantly improving our manuscript.
Reviewer 2 Report (Previous Reviewer 2)
The manuscript has been improved and my concerns to previous version were satisfied.
Please check the line#132: what is the Amicron variant?
Please reorganize your ref and numbers.
Author Response
Dear reviewer, thank you for your meticulous comments, which help us significantly improve our manuscript. We have edited the typo in line #132 and unified the reference list according to the MDPI citation guideline.
Reviewer 3 Report (Previous Reviewer 3)
Minor comments
I consider that the manuscript "SARS-CoV-2 Omicron (B.1.1.529) Variant: A challenge with 2 Covid-19" improved remarkably, I thank the authors for attending to the suggestions, and I only have two minor comments:
1. Line 132: Change “Amicron” to “Omicron”
2. Authors should carefully review the references and follow the journal format since there is currently no homogeneous format, some journal titles are abbreviated, and others are not.
Author Response
Dear reviewer, thank you for your meticulous comments, which help us significantly improve our manuscript. We have edited the typo in line #132 and unified the reference list according to the MDPI citation guideline.
This manuscript is a resubmission of an earlier submission. The following is a list of the peer review reports and author responses from that submission.
Round 1
Reviewer 1 Report
Title: SARS-CoV-2 Omicron (B.1.1.529) Variant: A challenge with COVID-19
Journal: Diagnostic journal
The authors did the thorough review on the topic highlighting the history of SARS-CoV-2 Omicron variant, its genomic features, pathogenic aspect, clinical aspect, and therapeutic aspect. I would like to give comments as below.
1. The abstract should express the brief summary of review result, its importance, impact on clinical and pathogenicity and the barriers (if present) and further suggestion of the authors in combating COVID-19 specifically SARS-CoV-2 Omicron variant from any aspect (public health or epidemiological point of view).
2. The authors should readjust the different fonts throughout the paper.
3. As for the review on mutation spectrum of Omicron variant, the authors focused on spike region only, although there are reports on some prominent mutation (eg. R203K+G204R double substitution mutation) found in nucleocapsid gene enhancing the viral transcription, replication and evading the immunity.
4. I suggested to list up the mutation of Omicron by region stating their functional changes, impact on transmission and replication or pathogenicity, with their references in table form.
5. In page 3 line 113, the authors stated that the Omicron variant is split into two sub-lineages: BA.1 and BA.2. Until now, there are main 5 sub-lineages of Omicron had been reported BA.1, BA.2, BA.3, BA.4 and BA.5 with newly emerged BA.2.75.2 threatening the immune invasion and resistant to currently available mRNA vaccine. The authors should consider discussing more detail on it.
6. As for the management of Omicron variant, the authors did not discuss about the role of convalescent plasma therapy if there are any reports.
7. There are many reports either on breakthrough infection or reinfection and the efficacy of currently available COVID-19 vaccine over Omicron variant prevention, the authors should concisely discuss based on the facts and the reported statistical values. 8. In the prevention section, although the vaccination is key role for prevention of COVID-19 or reducing the disease severity and mortality, the personal protection and awareness also plays an important role. The authors should discuss precisely on it.
Author Response
The authors would like to thank the editorial office and all the reviewers for their valuable comments and suggestions. We will try to meet the reviewers’ suggestions by extending and improving the investigation’s description as much as possible. Below we address each reviewer’s comments and indicate intended changes to the manuscript.
1- The abstract should express the brief summary of review result, its importance, impact on clinical and pathogenicity and the barriers (if present) and further suggestion of the authors in combating COVID-19 specifically SARS-CoV-2 Omicron variant from any aspect (public health or epidemiological point of view).
Response to point #1
Many thanks for your comment. The abstract is written to incorporate your suggestions.
2- The authors should readjust the different fonts throughout the paper.
Response to point #2
Many thanks for your comment. The fonts are adjusted throughout the manuscript.
3- As for the review on mutation spectrum of Omicron variant, the authors focused on spike region only, although there are reports on some prominent mutation (eg. R203K+G204R double substitution mutation) found in nucleocapsid gene enhancing the viral transcription, replication and evading the immunity.
Response to point #3
Many thanks for your comment. Other prominent mutations of the virus are added to the text.
4- I suggested to list up the mutation of Omicron by region stating their functional changes, impact on transmission and replication or pathogenicity, with their references in table form.
Response to point #4
Many thanks for your point. The information on different Omicron mutations is provided in the manuscript per your valuable comment.
5- In page 3 line 113, the authors stated that the Omicron variant is split into two sub-lineages: BA.1 and BA.2. Until now, there are main 5 sub-lineages of Omicron had been reported BA.1, BA.2, BA.3, BA.4 and BA.5 with newly emerged BA.2.75.2 threatening the immune invasion and resistant to currently available mRNA vaccine. The authors should consider discussing more detail on it.
Response to point #5
Many thanks for your point. We rewrote the statement mentioned above, considering new subvariants. We also added a new section in which we explained the features of these subvariants.
6- As for the management of Omicron variant, the authors did not discuss about the role of convalescent plasma therapy if there are any reports.
Response to point #6
Many thanks for your point. This topic is added to the manuscript.
7- There are many reports either on breakthrough infection or reinfection and the efficacy of currently available COVID-19 vaccine over Omicron variant prevention, the authors should concisely discuss based on the facts and the reported statistical values.
Response to point #7
Many thanks for your comment. A new section is added to the manuscript dedicated to the effectiveness of neutralizing antibodies and vaccines on this variant.
8- In the prevention section, although the vaccination is key role for prevention of COVID-19 or reducing the disease severity and mortality, the personal protection and awareness also plays an important role. The authors should discuss precisely on it.
Response to point #8
Many thanks for your comment. This section has been edited and rewritten to cover the subject you mentioned.
Reviewer 2 Report
The current draft briefly explaining the common features of the Omicron variant of Sars-Cov-2 from genome organization to disease prevention. Although there are some similar review articles out there, the current submitted review has some merit to publication. Please see my minor comments below,
Line # 28-29: Remove the sentence starting with “Previous research…” from abstract, It`s not relevant in between. You might consider it shifting to right before the last sentence of the abstract that “This review summarizes….”. Even so, please don`t use “previous research”, you can use accumulating studies, mounting evidences or several reports instead.
Line#32: Please remove “those with”.
Line#54-56: Please rewrite this last statement since it`s similar with the one at line 36-39.
Line#69-71: I`m not sure this sentence “Letters and numbers are ……virus protein [11].” is required here, please remove.
Line #78: replace “due to” with “with” and “in the” with “from”.
Line#127: No need to abbreviate the “RBM”, same for BFE at #136. Please go through your draft and make sure you only abbreviated the words that are used at least twice.
Line #128: You already abbreviated the “ACE2” before, no need to spell it out. Same for CT at #222, and for WHO at #50 and #64, SARS-Cov-2 at #49, Covid-19 at #58, RBD at # 81, ICU at #335. There might be many more, please correct.
Line #268: please capitalize the letter of the protein “S”.
Author Response
The authors would like to thank the editorial office and all the reviewers for their valuable comments and suggestions. We will try to meet the reviewers’ suggestions by extending and improving the investigation’s description as much as possible. Below we address each reviewer’s comments and indicate intended changes to the manuscript.
1- Line # 28-29: Remove the sentence starting with “Previous research…” from abstract, It`s not relevant in between. You might consider it shifting to right before the last sentence of the abstract that “This review summarizes….”. Even so, please don`t use “previous research”, you can use accumulating studies, mounting evidences or several reports instead.
Response to point #1
Many thanks for your comment. We edited the abstract according to your comment.
2- Line#32: Please remove “those with”.
Response to point #2
Many thanks for your comment. We removed the above words.
3- Line#54-56: Please rewrite this last statement since it`s similar with the one at line 36-39.
Response to point #3
Many thanks for your comment. We rewrote the statement mentioned above.
4- Line#69-71: I`m not sure this sentence “Letters and numbers are ……virus protein [11].” is required here, please remove.
Response to point #4
Many thanks for your comment. We acknowledged that this sentence is not required at that part, and it is omitted.
5- Line #78: replace “due to” with “with” and “in the” with “from”.
Response to point #5
Many thanks for your comment. We replaced the words mentioned above with your suggestions.
6- Line#127: No need to abbreviate the “RBM”, same for BFE at #136. Please go through your draft and make sure you only abbreviated the words that are used at least twice.
Response to point #6
Many thanks for your comment. We omitted these abbreviations and also rechecked the whole manuscript.
7- Line #128: You already abbreviated the “ACE2” before, no need to spell it out. Same for CT at #222, and for WHO at #50 and #64, SARS-Cov-2 at #49, Covid-19 at #58, RBD at # 81, ICU at #335. There might be many more, please correct.
Response to point #7
Many thanks for your comment. We omitted these abbreviations and also rechecked the whole manuscript.
8- Line #268: please capitalize the letter of the protein “S”.
Response to point #8
Many thanks for your comment. We capitalized the letter of the protein “S”.
Reviewer 3 Report
General comment
The manuscript "SARS-CoV-2 Omicron (B.1.1.529) Variant: A challenge with 2 Covid-19" is well structured, clear, and consistent with the article's title. The authors make a good review of essential aspects known to date about the Omicron variant, including its history, genome, transmissibility, clinical manifestations, diagnosis, management, and effectiveness of existing vaccines against this pathogen. However, I have some suggestions and questions that, from my point of view, could improve this work.
Comments
Although it seems like a good revision, I suggest including a section on “Sublineages of the Omicron variant” mentioning its importance and repercussions on the host.
Globally and in the Americas region, how is the distribution of sublineages?
Is the behavior of the Omicron variant sublineages known in terms of transmissibility, clinical symptoms, severity, or evasion of the immune response?
References
Authors should carefully review the references and follow the journal's format, as there is currently no homogeneous format.
Author Response
The authors would like to thank the editorial office and all the reviewers for their valuable comments and suggestions. We will try to meet the reviewers’ suggestions by extending and improving the investigation’s description as much as possible. Below we address each reviewer’s comments and indicate intended changes to the manuscript.
1- Although it seems like a good revision, I suggest including a section on “Sublineages of the Omicron variant” mentioning its importance and repercussions on the host.
Response to point #1
Many thanks for your comment. we included a new section providing more information on new subvariants of the Omicron.
2- Globally and in the Americas region, how is the distribution of sublineages?
Response to point #2
Many thanks for your comment. We discussed your suggestions in the section mentioned above.
3- Is the behavior of the Omicron variant sublineages known in terms of transmissibility, clinical symptoms, severity, or evasion of the immune response?
Response to point #3
Many thanks for your comment. It was challenging to precisely identify the transmissibility, clinical symptoms, severity, or evasion of the immune response of the new subvariants because there were not enough papers about them. However, we provided a table in the new section and tried to assess transmissibility, clinical symptoms, severity, or evasion of the immune response based on mutations.
4- Authors should carefully review the references and follow the journal's format, as there is currently no homogeneous format.
Response to point #3
Many thanks for your comment. The references are updated based on the journal's format.